# A High-Fat Diet Induces Muscle Mitochondrial Dysfunction and Impairs Swimming Capacity in Zebrafish: A New Model of Sarcopenic Obesity

**DOI:** 10.3390/nu14091975

**Published:** 2022-05-09

**Authors:** Yun-Yi Zou, Zhang-Lin Chen, Chen-Chen Sun, Dong Yang, Zuo-Qiong Zhou, Qin Xiao, Xi-Yang Peng, Chang-Fa Tang

**Affiliations:** Key Laboratory of Physical Fitness and Exercise Rehabilitation of Hunan Province, College of Physical Education, Hunan Normal University, Changsha 410012, China; 202020151346@hunnu.edu.cn (Y.-Y.Z.); zhanglinchen@hunnu.edu.cn (Z.-L.C.); sunchenchen1022@hunnu.edu.cn (C.-C.S.); yangdong@hunnu.edu.cn (D.Y.); zhouzuoqiong@hunnu.edu.cn (Z.-Q.Z.); qinzixiao715@163.com (Q.X.)

**Keywords:** high-fat diet, muscle, sarcopenic obesity, mitochondria, zebrafish

## Abstract

Obesity is a highly prevalent disease that can induce metabolic syndrome and is associated with a greater risk of muscular atrophy. Mitochondria play central roles in regulating the physiological metabolism of skeletal muscle; however, whether a decreased mitochondrial function is associated with impaired muscle function is unclear. In this study, we evaluated the effects of a high-fat diet on muscle mitochondrial function in a zebrafish model of sarcopenic obesity (SOB). In SOB zebrafish, a significant decrease in exercise capacity and skeletal muscle fiber cross-sectional area was detected, accompanied by high expression of the atrophy-related markers Atrogin-1 and muscle RING-finger protein-1. Zebrafish with SOB exhibited inhibition of mitochondrial biogenesis and fatty acid oxidation as well as disruption of mitochondrial fusion and fission in atrophic muscle. Thus, our findings showed that muscle atrophy was associated with SOB-induced mitochondrial dysfunction. Overall, these results showed that the SOB zebrafish model established in this study may provide new insights into the development of therapeutic strategies to manage mitochondria-related muscular atrophy.

## 1. Introduction

Obesity is an epidemic disease in economically developed and developing countries and is associated with several chronic diseases, including diabetes mellitus and non-alcoholic fatty liver disease [1]. Unhealthy diets, such as a high-fat diet (HFD), are major causes of obesity [2,3]. Furthermore, HFDs can also contribute to metabolic disorders, pro-inflammatory cytokine expression, and insulin resistance in skeletal muscle [4], the latter of which is one of the primary causes of type 2 diabetes mellitus [5,6]. Long-term, high-fat dietary intake can induce myofibrillar protein degradation and myonuclear apoptosis and ultimately lead to sarcopenia [7]. Sarcopenic obesity (SOB) combined with the deterioration of skeletal muscle function can lead to decreased exercise capacity, accompanied by increased disability and frailty [8]. Thus, further studies of the mechanisms underlying the pathogenesis of SOB are necessary to facilitate the prevention and treatment of this muscle disorder.

Mitochondria are crucial organelles responsible for the ATP supply and metabolic status of skeletal muscle [9]. The reduction in muscle mass induced by SOB may contribute to dysregulation of the mitochondrial network and impairment of mitochondrial function [10,11]. Additionally, the dysregulation of mitochondrial function can accelerate lipid deposition and oxidative stress in muscle, thereby exacerbating the decrease in muscle mass [12]. Thus, the investigation of the molecular mechanisms regulating mitochondrial function and how obesity leads to mitochondrial dysfunction in sarcopenia may facilitate the development of strategies to manage SOB.

Many researchers have developed reliable zebrafish models to understand muscle degeneration and its pathophysiology [13,14]. We hypothesized that zebrafish could be an appropriate SOB model and that the molecular mechanism underlying the condition was related to mitochondrial function. In the current study, we evaluated the effects of HFD-induced SOB on muscle atrophy and exercise capacity in a zebrafish model. We also explored the causal relationship between muscle mitochondrial function and SOB in zebrafish. Our findings may provide insights into therapeutic targets and treatment strategies for SOB.

## 2. Materials and Methods

### 2.1. Animal Ethics Statement

All experiments were conducted under the Chinese guidelines for animal welfare and experimental protocols. Approval was obtained from the Animal Experiment Administration Committee of Hunan Normal University (Changsha, China; approval number: 2018/046).

### 2.2. Animals, Culture Conditions, and Dietary Experimental Design

Adult male zebrafish (AB strain, 4 months old) were raised under light at 28 °C for 14 h under standard husbandry conditions. Prior to the start of the experiment, zebrafish were fed a commercial diet (TP1FM21051; Trophic Animal Feed High-Tech Co., Ltd., Nantong, China) containing 6% fat for 1 week of adaptation. Subsequently, 90 healthy zebrafish were selected and randomly allocated to two diet groups (three tanks per dietary treatment, 15 zebrafish per tank): a normal diet (6% fat; ND) or an HFD (24% fat). During the 16-week feeding trial, experimental zebrafish were fed three times per day. At the end of the trial, the final body weight and body length of the fish in each tank were determined. Fifteen zebrafish from each dietary regimen group (5 fish/tank) were randomly selected for quantification of swimming capacity. Subsequently, zebrafish were randomly anesthetized with 20 mg/L tricaine methanesulfonate after fasting for 24 h, after which blood samples were collected from the caudal vein using 10 μL heparin-treated capillaries for blood glucose measurement. Liver and muscle samples were immediately collected for further experiments.

### 2.3. Biochemical Analyses

Blood glucose levels were measured using a blood glucometer purchased from Yuwell (Yuwell 580, Beijing, China). For measurement of muscle triglyceride, total cholesterol, catalase, and total superoxide dismutase levels, 150 mg of muscle tissue (from three zebrafish) from each group was homogenized in 0.9% normal saline (1:9, *v*/*v*) and centrifuged at 2000 rpm for 15 min. The supernatants were collected and aliquoted for further analysis. Biochemical parameters were measured using specific commercial kits (Jiancheng Biotech Co., Nanjing, China).

### 2.4. Histological Analysis

Liver samples from three zebrafish per group (one zebrafish per tank) were fixed in 4% paraformaldehyde solution for 24 h, embedded in paraffin, and sliced into 4-µm-thick sections for hematoxylin-eosin (H&E) staining and Masson staining.

Three fresh liver samples were fixed in 4% paraformaldehyde for 4 h and embedded in OCT. Frozen sections (8 µm thickness) were generated using a cryostat and the samples were fixed with 4% paraformaldehyde for an additional 30 min. The slides were washed in distilled water and stained with Oil Red O for 15 min. Oil Red O staining was performed to determine the lipid contents in liver tissue.

The muscle tissues (for H&E staining) of zebrafish (*n* = 3) were fixed in 4% paraformaldehyde, dehydrated through an ethanol series, dewaxed with xylene, and embedded in paraffin wax. The samples were transversely cut into 4 μm thick slices, dewaxed, dried, and stained with H&E under standard conditions. The images were captured using a microscope (Leica, Heidelberg, Germany) and analyzed using ImageJ (NIH, Bethesda, MD, USA).

### 2.5. Transmission Electron Microscopy

For transmission electron microscopy, tissues were fixed in 2.5% glutaraldehyde for 6–12 h. The fixative solution was discarded, and cells were transferred to phosphate-buffered saline. Next, cells were fixed in 1% osmic acid for 1–2 h, and dehydration was carried out by incubation in 30% ethanol for 10 min, 50% ethanol for 10 min, 70% uranyl acetate in ethanol (stained before embedding) for 3 h or overnight, 80% ethanol for 10 min, 95% ethanol for 15 min, 100% ethanol twice for 50 min each, and propylene oxide for 30 min. Next, samples were incubated in propylene oxide: epoxy resin (1:1) for 1–2 h and then in pure epoxy resin for 2–3 h. After embedding in pure epoxy resin, samples were baked in an oven at 40 °C for 12 h and then at 60 °C for 48 h. Samples were then cut into ultra-thin sections and placed on copper grids. Staining was then performed with lead and uranium stain, and images were acquired using a Japan Electronics JEM1400 transmission electron microscope and recorded with a Morada G3 digital camera.

### 2.6. Swimming Capacity and Oxygen Consumption Measurement

Analysis of the swimming performance and oxygen consumption of zebrafish was performed using a miniature swimming tunnel respirator (Loligo Systems, Viborg, Denmark). The following formula was used to calculate U_crit_ values for the swimming tests: U_crit_ = Uf + US × (Tf/TS), where Uf (cm/s) is the highest velocity, US (2.7 L) is the velocity increment, Tf (min) is the time elapsed at fatigue velocity, and TS (14 min) is the prescribed interval time. U_crit_ is expressed in terms of body lengths per second (BL/s). Maximal oxygen consumption (MO_2_) was calculated using AutoResp 1 software (Loligo Systems, Viborg, Denmark). More detailed information can be found in our previous study [15].

### 2.7. Total RNA Extraction and Reverse Transcription-Quantitative Polymerase Chain Reaction (RT-qPCR)

The total RNA from zebrafish muscle tissue (one sample with two fish muscle tissues, *n* = 6) was extracted by homogenization in TRIzol solution (Thermo Fisher Scientific, Waltham, MA, USA) according to the manufacturer′s protocol. RNA was reverse-transcribed into cDNA using a reverse transcription system kit (Takara, Tokyo, Japan). qPCR was performed using SYBR Green Master Mix (Thermo Fisher Scientific). Relative mRNA expression was determined using a Bio-Rad real-time PCR system (CFX96; Bio-Rad Laboratories, Hercules, CA, USA). Sangon Biotech synthesized primers for the detected genes and the reference gene, *gapdh*. Relative mRNA expression was determined using the 2^−ΔΔCT^ method.

### 2.8. Western Blot

Zebrafish skeletal muscle tissue (*n* = 6) was lysed in cold RIPA buffer (Solarbio, Beijing, China) containing a mixture of protease and phosphatase inhibitors (Solarbio). Protein quantification was performed using a bicinchoninic acid protein assay kit (cat. no. E112-01/02; Vazyme, Nanjing, China), and the protein (30 mg) in each sample was separated by sodium dodecyl sulfate-polyacrylamide gel electrophoresis on 10% or 15% gels and then transferred to 0.45 or 0.22 mm polyvinylidene difluoride (PVDF) membranes. For analysis of phosphorylated proteins, membranes were blocked with 5% bovine serum albumin, whereas for analysis of other proteins, membranes were blocked with 5% fat-free milk. Membranes were incubated with primary antibodies overnight at 4 °C and washed with TBST. Subsequently, membranes were incubated with appropriate secondary horseradish peroxidase-conjugated antibodies. The proteins were detected using a gel imaging system (Tanon, Shanghai, China). The antibodies used were as follows: rabbit anti-β-actin antibody (1:2000; Proteintech, Wuhan, China), rabbit anti-Atrogin-1 antibody (1:1000; Proteintech), rabbit anti-muscle RING-finger protein-1 (MuRF1) antibody (1:1000; Proteintech), rabbit anti-sirtuin 1 (SIRT1; 1:1500; Proteintech), rabbit anti-peroxisome proliferator-activated receptor γ coactivator 1-α (PGC1α) antibody (1:1000; Bioss, Beijing, China), rabbit anti-optic atrophy protein 1 (OPA1) antibody (1:1500; Proteintech), rabbit anti-mitofusin 2 (MFN2) antibody (1:1500; Proteintech), rabbit anti-dynamin-related protein 1 (DRP1) antibody (1:1500; Proteintech), and rabbit anti-phospho-AMP-activated protein kinase (AMPK; Thr172) antibody (1:2000; Cell Signaling Technology, Danvers, MA, USA). Protein expression was normalized to that of β-actin.

### 2.9. Statistical Analysis

Statistical analysis was performed using SPSS (version 22.0; IBM, Chicago, IL, USA) or the GraphPad Prism software (version 9.0; San Diego, CA, USA). Data were expressed as means ± standard deviations of three independent experiments. Unpaired *t*-tests were used to compare the mean values of the two groups. All experiments were repeated three times. Statistical significance was set at *p* < 0.05.

## 3. Results

### 3.1. Long-Term HFD Feeding Induced Obesity and Liver Injury in Zebrafish

As shown in Figure 1, zebrafish fed an HFD exhibited higher body weight compared with those in the ND group (Figure 1A), whereas no significant differences in body length were observed (Figure 1B). Fasting blood glucose and muscle triglyceride contents significantly increased in the HFD group (Figure 1C,D). After 16 weeks, HFD-fed zebrafish developed the characteristics of fatty liver with evident increases in lipid droplets, hepatic steatosis severity, and collagen contents (Figure 1E). These data indicate the successful establishment of HFD-induced obesity in zebrafish.

### 3.2. Long-Term HFD Feeding Induced Skeletal Muscle Atrophy

After a 16-week feeding trial, we used transmission electron microscopy to examine the fiber morphology of muscles. Compared with the ND group, in which zebrafish muscle exhibited a normal structure, the muscles of zebrafish in the HFD group exhibited more lipid droplets. H&E staining indicated that the muscle fiber sizes of transverse sections in the HFD group were smaller than those in the ND group (Figure 2A,B). Furthermore, the expression of muscle atrophy markers (Atrogin-1 and MuRF1) was significantly increased in zebrafish in the HFD group (Figure 2C,D). These results demonstrated that long-term HFD feeding caused muscular atrophy in zebrafish.

### 3.3. Long-Term HFD Feeding Impaired the Swimming Capacity of Zebrafish

We used exercise experimental protocols to measure the swimming capacity of the two groups of zebrafish. As shown in Figure 3A,B, as the water sped up beyond 6.2 BL/s, zebrafish in the ND group exhibited higher swimming capacity and MO_2_ levels than those in the HFD group at each testing stage. Moreover, zebrafish in the HFD group exhibited decreased exhaustive swimming times and MO_2max_ values compared with those in the ND group (Figure 3C,D). Furthermore, zebrafish in the HFD group exhibited slower U_crit_ and U_crit-r_ when compared with those in the ND group (Figure 3E,F). These results indicated that the swimming capacity of HFD-fed zebrafish was impaired.

### 3.4. Long-Term HFD Feeding Suppressed Skeletal Muscle Mitochondrial Biogenesis and Fatty Acid Oxidation-Related Gene Expression in Zebrafish

We studied the biogenesis and oxidation functions of muscle mitochondria to elucidate the potentially damaging effects of HFD feeding. AMPK, SIRT1, and PGC1α maintain normal mitochondrial biogenesis and fatty acid oxidation in skeletal muscle. As expected, we observed lower levels of phospho-AMPK, SIRT1, and PGC1α in the muscles of HFD-fed zebrafish compared with those in zebrafish in the ND group (Figure 4A,B). Moreover, we detected significant decreases in the expression of genes related to mitochondrial biogenesis (*pgc1α*, *nrf1*, and *tfam*), fatty oxidation (*pparab* and *cpt1a*), and electron transport chain (ETC) complex subunits (*sdha*, *uqcrc2b*, *cox4il*, and *atp5f*; Figure 4C). These data showed that HFD feeding may partially show a suppression of mitochondrial function.

### 3.5. Long-Term HFD Feeding Induced Abnormal Mitochondrial Fusion and Fission in Zebrafish Skeletal Muscle

Abnormal mitochondrial fusion and fission are responsible for the impairment of mitochondrial function [16]. Compared with the ND group, skeletal muscles of zebrafish in the HFD group exhibited higher expression levels of mitochondrial fission-related proteins (e.g., DRP1) and lower levels of mitochondrial fusion-related proteins (e.g., OPA1 and MFN2; Figure 5A,B). Moreover, as shown in Figure 5C, most mitochondria in the HFD group were degenerated, enlarged, and swollen, and the mitochondrial cristae were broken or absent. These data suggested that long-term HFD feeding could promote mitochondrial fission and suppress mitochondrial fusion.

## 4. Discussion

Obesity is an epidemic condition [17]. Severe obesity is accompanied by excessive accumulation of fat in visceral adipose tissue and normally lean tissues (e.g., the liver, heart, and skeletal muscle) [18]. Lipid overload in the skeletal muscle lowers muscle mass and induces sarcopenia [19,20]. Zebrafish is a promising model system for studying human diseases, such as obesity and sarcopenia, because of the functional conservation in substance and energy metabolism and the similar characteristics in muscular physiology. In this study, we fed adult zebrafish an HFD for 16 weeks to induce SOB and showed that long-term HFD feeding contributed to muscular atrophy and decreased swimming capacity in obese zebrafish. Moreover, we demonstrated that this outcome was related to mitochondrial dysfunction in the muscles of SOB-model zebrafish.

Obesity is primarily caused by the consumption of an unhealthy diet with a prolonged imbalance of energy intake and energy expenditure Here, we observed a clear increase in body weight and muscle triglyceride content in zebrafish in the HFD group. Additionally, zebrafish fed an HFD exhibited typical characteristics of fatty liver. These results indicated that zebrafish developed obesity after 16 weeks of HFD feeding. Glucose disposal activated by insulin is a vital metabolic function of skeletal muscle, and normal glucose metabolism is important for health [21]. Moreover, excessive fat accumulation in skeletal muscles can impair insulin signaling and glucose intake [22]. Most obese phenotypes are accompanied by the inhibition of insulin receptor substrate and AKT and increased insulin resistance [23]. The current findings further confirmed that long-term HFD feeding contributed to obesity in zebrafish.

SOB has become a subject of interest and research over the years [24]. SOB represents a combination between sarcopenia and obesity, and there are several common pathophysiological mechanisms between sarcopenia and obesity, resulting in increased disease severity when both conditions are present [19]. Skeletal muscle fiber atrophy is the major pathological trait observed in sarcopenia. In our study, we found that skeletal muscle fiber size was decreased in obese zebrafish and that the expression levels of muscle atrophy marker proteins (Atrogin-1 and MuRF1) were significantly elevated in obese skeletal muscles. These findings indicate that increased protein degradation and muscle atrophy were induced by HFD feeding in zebrafish. Moreover, we detected an evident decrease in swimming capacity in SOB zebrafish, with clear reductions in exhaustive swimming time, MO_2max_, U_crit_, and U_crit-r_.

Mitochondria play essential roles in regulating fatty acid metabolism and ATP production, which are important for muscle contractibility and plasticity [25]. Impaired mitochondrial function in skeletal muscle is thought to promote muscular atrophy in subjects with obesity [10]. Indeed, a few weeks of HFD feeding can cause intramyocellular lipid accumulation and reduce mitochondrial function [26]. The AMPK/SIRT1 pathway maintains cellular energy stores in skeletal muscles [27]. In addition, as the main regulator of mitochondrial biogenesis, PGC1α can be stimulated by the AMPK/SIRT1 pathway [28]. The AMPK/SIRT1/PGC1α pathway is inhibited in the skeletal muscles of humans and mice with obesity [29,30]. These findings are also supported by our current results. In this study, we observed decreased levels of phospho-AMPK, SIRT1, and PGC1α proteins as well as downregulation of mRNA levels of various transcription factors (*nrf1* and *tfam*) in the muscles of SOB-model zebrafish. Moreover, mitochondrial biogenesis is responsible for oxidative phosphorylation and fatty acid β-oxidation [31]. This reduction in mitochondrial biogenesis may result in decreased ATP synthesis and fatty acid β-oxidation. In our study, we also observed the downregulation of genes involved in the mitochondrial complex and β-oxidation. These results indicated that skeletal muscle atrophy was associated with mitochondrial dysfunction.

Mitochondria are unique and highly dynamic organelles that exhibit continuous fission and fusion [32]. Fusion and fission processes regulate the length, size, and morphology of mitochondria, and these mitochondrial dynamics are associated with mitochondrial function. Under abnormal physiological conditions, mitochondria cannot maintain an appropriate balance between fission and fusion. Fusion ameliorates stress by mixing the contents of partially damaged mitochondria as a form of complementation, whereas fission is thought to induce oxidative stress and cell apoptosis [33]. Thus, mitochondrial homeostasis is critical for maintaining normal cellular events. MFN2 is responsible for outer membrane fusion, whereas OPA1 is required for inner membrane fusion. In the current study, we observed a strong inhibition of MFN and OPA1 in SOB model zebrafish. Moreover, we observed increased expression of the mitochondria fission marker DRP1, accompanied by abnormal mitochondrial structures in the HFD group. These findings indicated that a decreased mitochondrial quality occurred in the skeletal muscles and mitochondria of SOB model zebrafish.

In summary, in HFD-fed SOB model zebrafish, muscle atrophy was related to mitochondrial dysfunction. Considering the complex pathogenesis of SOB, such as lipotoxicity, mitochondrial dysfunction, inflammation, and insulin resistance, the current study only unveiled one aspect of it. Further comprehensive studies are necessary to investigate the detailed mechanisms linking the pathogenesis of SOB and mitochondria dysfunction and to assess the efficacy of pharmacological and exercise-based interventions targeting mitochondria to prevent or treat SOB.

## Figures and Tables

**Figure 1 nutrients-14-01975-f001:**
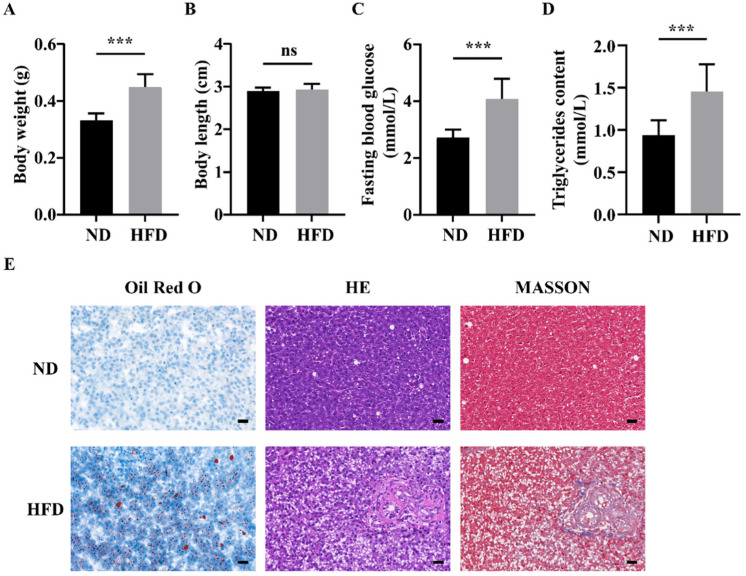
Long-term high-fat diet (HFD) feeding induced obesity and liver injury in zebrafish. (**A**) Body weight of zebrafish. (**B**) Body length of zebrafish. (**C**) Fasting blood glucose of zebrafish. (**D**) Muscle triglyceride contents. (**E**) Oil Red O staining, H&E staining, and Masson staining of zebrafish livers. ***, *p* < 0.001. Data represent means, and error bars represent standard errors of the means. Scale bar, 20 μm. ND, normal diet; HFD, high-fat diet.

**Figure 2 nutrients-14-01975-f002:**
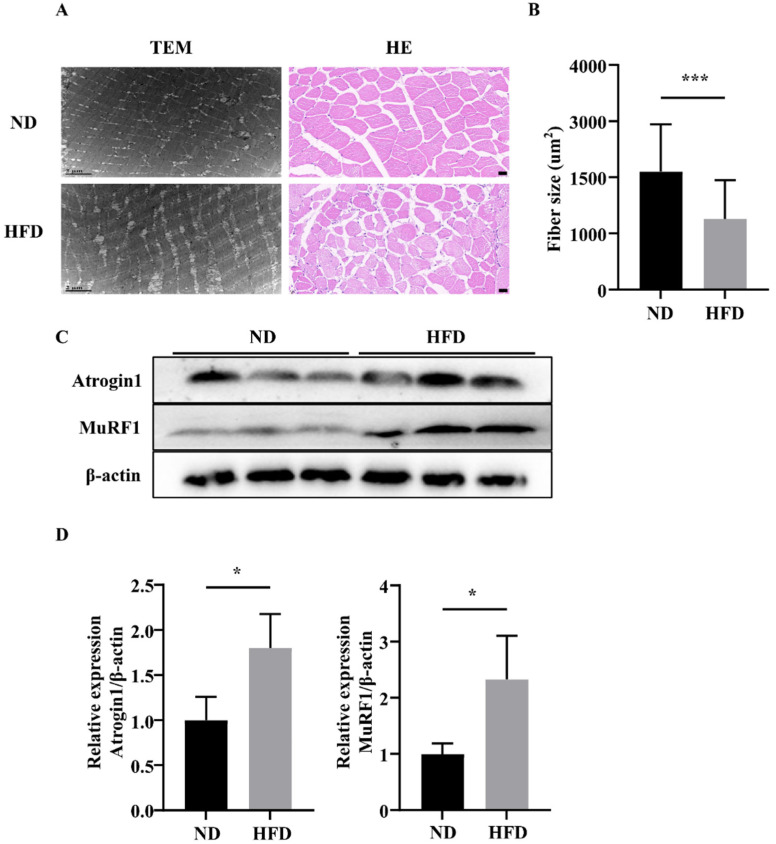
Comparison of skeletal muscle fiber size between zebrafish groups. (**A**) Representative photomicrographs of muscle sections stained H&E or imaged using transmission electron microscopy. (**B**) Average fiber size (based on H&E staining). (**C**,**D**) Atrogin-1 and MuRF1 protein expression. *, *p* < 0.05, ***, *p* < 0.001. Data represent means, and error bars represent standard errors of the means. Scale bars in transmission electron microscopy images, 2 μm. Scale bars in H&E-stained images, 20 μm. ND, normal diet; HFD, high-fat diet.

**Figure 3 nutrients-14-01975-f003:**
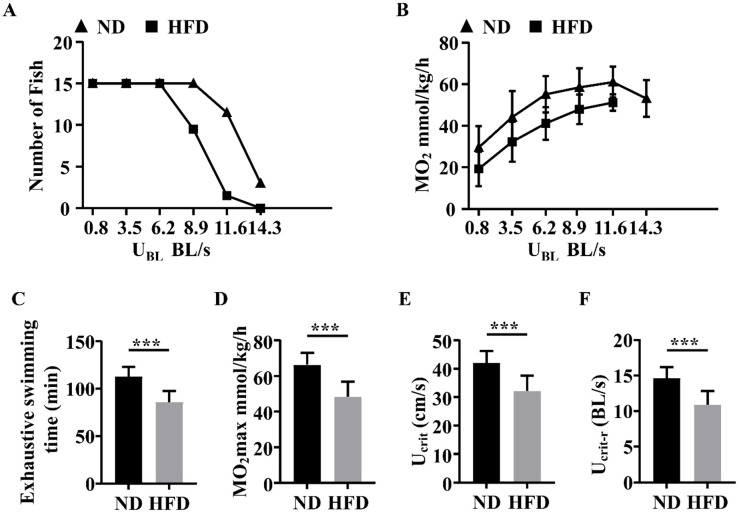
Comparison of swimming capacity tests between the two groups of zebrafish. (**A**) The number of zebrafish in the two groups at each speed test stage. (**B**) MO_2_ levels in the two groups at each testing stage. (**C**) Exhaustive swimming times of zebrafish. (**D**) MO_2max_ of zebrafish. (**E**) U_crit_ of zebrafish. (**F**) U_crit-r_ of zebrafish. ***, *p* < 0.001. Data represent means, and error bars represent standard errors of the means. Scale bar, 20 μm. ND, normal diet; HFD, high-fat diet; U_crit_, critical swimming speed; MO_2_, oxygen consumption.

**Figure 4 nutrients-14-01975-f004:**
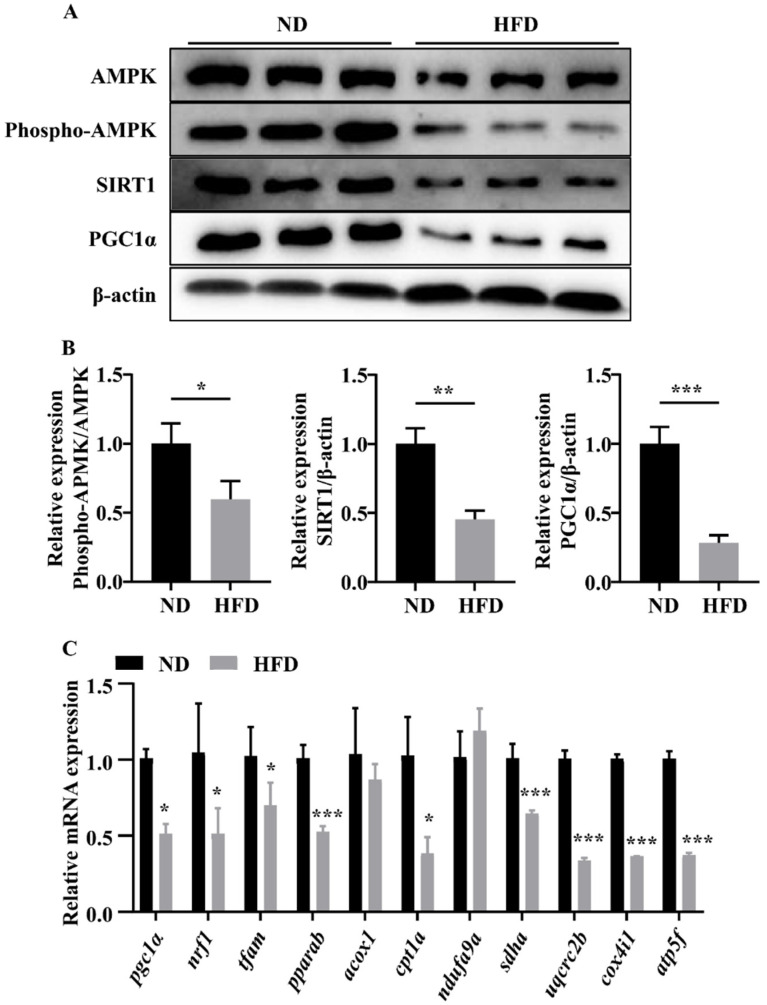
Long-term HFD feeding induced abnormal mitochondrial dysfunction. (**A**,**B**) Phospho-AMPK, SIRT1, and PGC1α protein levels. (**C**) mRNA expression of genes related to mitochondrial biogenesis, fatty oxidation, and ETC complexes subunits. *, *p* < 0.05, **, *p* < 0.01, ***, *p* < 0.001. Data represent means, and error bars represent standard errors of the means. ND, normal diet; HFD, high-fat diet; ETC, electron transport chain.

**Figure 5 nutrients-14-01975-f005:**
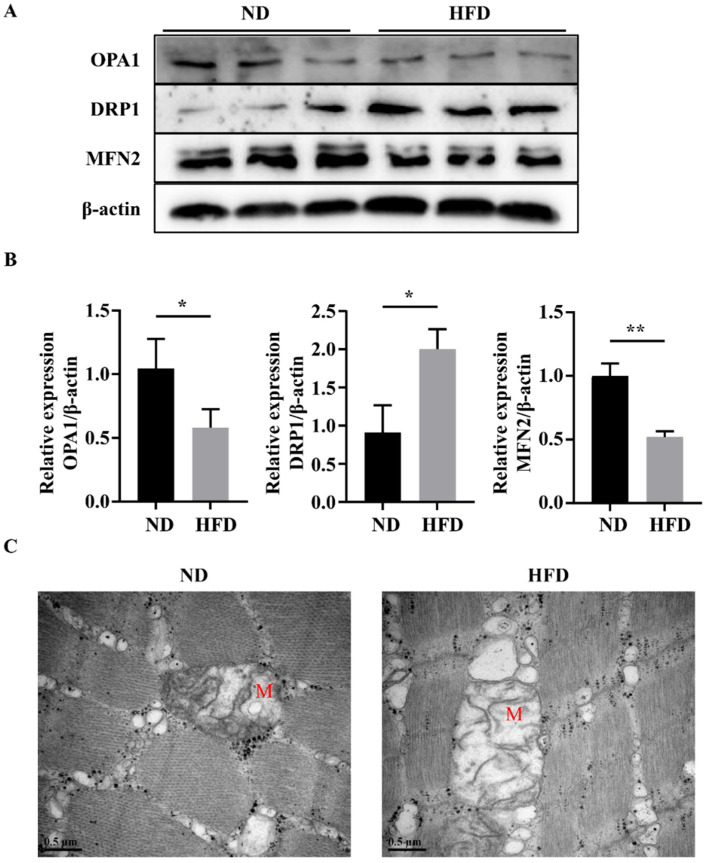
Long-term HFD feeding induced abnormal mitochondrial fusion and fission. (**A**,**B**) Protein expression of OPA1, MFN2, and DRP1. (**C**) Representative photomicrographs of muscle sections imaged using transmission electron microscopy. M, mitochondria. *, *p* < 0.05, **, *p* < 0.01. Data represent means, and error bars represent standard errors of the means. Scale bar in transmission electron microscopy images, 0.5 μm. ND, normal diet; HFD, high-fat diet.

## Data Availability

The data are available from the corresponding author.

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
