# Peer review of "A High-Fat Diet Induces Muscle Mitochondrial Dysfunction and Impairs Swimming Capacity in Zebrafish: A New Model of Sarcopenic Obesity"

_nutrients, 2022, doi:10.3390/nu14091975_

Round 1

Reviewer 1 Report

Authors investigated the effect of a high fat diet on muscle atrophy, mitochondria dysfunction, and \ exercise capacity in a zebrafish model of sarcopenia obesity. Though the methodology and experimental techniques utilized were well conducted, this reviewer questions the study’s contribution to the field, and its appropriateness to be included in Nutrients. As framed, the manuscript suggests these findings are novel and provide biological insights which can be used to develop therapeutic interventions. However, the effects of a high-fat diet on multiple muscle outcomes have already been reported in the literature, both in human and rodent models.

Thus, this reviewer questions the author’s conclusion, “Overall, these results provide a pharmacological and biological foundation for the development of therapeutic strategies to manage mitochondria-related muscle atrophy”, and whether the current study contributes significantly to the literature.

Provide justification for using the Zebrafish model vs. a rodent model. What can be yielded from this model versus other models for the project’s specific question.

Authors claim “...results provide a pharmacological foundation for the development of therapeutic strategies…: . Provie justification for this claim.

How can these results be used to develop therapeutic strategies?

Other edits:

Line 14: Correction: “…however, whether decreased mitochondrial…” Remove the “a”

Line 17: Change obese to SOB zebrafish. As written, appears there are two separate models, an Obese model and a SOB model. Clarify.

Line 22-23: How do your results support such a claim? What pharmacological foundation do these results provide? Based upon the biological results, this conclusion is overreaching.

Line 32-33: Missing the word “High” or something equivalent in sentence “Long-term….dietary fat intake”

This reviewer recommends additional grammar and paragraph structure assistance when finalizing the manuscript. Though written clearly, paragraphs feel disjointed and consistent. 

Reviewer 2 Report

Reference 1 is from 2004 – there are many more relevant and recent reviews. One suggestion is: Blüher, M. Obesity: Global epidemiology and pathogenesis. Nat. Rev. Endocrinol. 2019, 15, 288-298, doi:10.1038/s41574-019-0176-8.

References 23-25 are from 2009 – are these still most appropriate references for this rapidly-advancing topic?

The Introduction does not give any references on zebrafish as a model of muscle disease in humans. One suggestion would be: Daya, Donaka, Karasik: Zebrafish models of sarcopenia. Dis Model Mech. 2020 Mar 30;13(3): doi: 10.1242/dmm.042689. 

Why were zebrafish chosen for this study? This needs to be clearly explained in the Introduction.

What is the hypothesis for this study?

Line 68: define MS-222 as tricaine methanesulphonate.

Lines 206-207 (and lines 275-276 and 291-292): Significant changes in mitochondrial biogenesis and histology have been shown but this statement that mitochondrial function was suppressed has not been shown by appropriate functional data.

Please include a discussion of other literature in zebrafish and its applicability to chronic human diseases, especially obesity and sarcopenia. 

Does reversal of obesity also reverse sarcopenia in humans and in this model? If this relationship occurs, is it causal or casual?

Reviewer 3 Report

Its a nice study reporting the relationship between obesity and muscle mitochondrial function in zebra fish model of obesity. The introduction is nicely written, results are well presented and discussed. Methods section needs few missing details. Also in the discussion, it would be nice if author can compare their findings with the previous studies. There are few clarifications needed, see attached file.

Round 2

Reviewer 1 Report

Excellent revisions and a more appropriate conclusion based upon findings. Suggest changing title to reflect revised conclusion. One example: "A high-fat diet induces muscle mitochondrial dysfunction and impairs swimming capacity in zebrafish: A new model of sarcopenic obesity".